# Value-Added Utilization of Citrus Peels in Improving Functional Properties and Probiotic Viability of *Acidophilus-bifidus-thermophilus* (ABT)-Type Synbiotic Yoghurt during Cold Storage

**DOI:** 10.3390/foods11172677

**Published:** 2022-09-02

**Authors:** Hayam M. Fathy, Ahmed A. Abd El-Maksoud, Weiwei Cheng, Fouad M. F. Elshaghabee

**Affiliations:** 1Microbiology Department, Faculty of Agriculture, Cairo University, Giza 12613, Egypt; 2Dairy Science Department, Faculty of Agriculture, Cairo University, Giza 12613, Egypt; 3Institute for Innovative Development of Food Industry, Institute for Advanced Study, Shenzhen University, 3688 Nanhai Road, Nanshan District, Shenzhen 518060, China

**Keywords:** citrus peels, synbiotic yoghurt, polyphenols, antioxidant, antibacterial, probiotic viability

## Abstract

Citrus peel, a fruit-processing waste, is a substantial source of naturally occurring health-promoting compounds, including polyphenols, and has great potential as a dietary supplement for enhancing the functional properties of food. The present work aimed to investigate the effects of sour orange (SO), sweet orange (SWO), and lemon (LO) peels on the typical physiochemical, antioxidant, antibacterial, and probiotic properties of synbiotic yoghurt fermented by *acidophilus-bifidus-thermophilus* (ABT)-type cultures during cold storage (0–28 days). High-performance liquid chromatography-diode array detection (HPLC-DAD) analysis showed that the total phenolic content in the SO peel were more than 2-fold higher than that in the SWO and LO peel. The predominant phenolic compounds were myricetin (2.10 mg/g dry weight) and *o*-coumaric acid (1.13 mg/g) in SO peel, benzoic acid (0.81 mg/g) and naringin (0.72 mg/g) in SWO peel, and benzoic acid (0.76 mg/g) and quercetin (0.36 mg/g) in LO peel. Only 0.5% (*w*/*w*) of citrus peel addition did not reduce the overall acceptance of ABT synbiotic yoghurt but led to increased acidity and decreased moisture during cold storage (14 and 28 days). Additionally, compared to control samples without citrus peel addition, supplementation with citrus peels improved the antioxidant property of the ABT synbiotic yoghurt. ABT milks with SO and SWO peel addition had significantly stronger DPPH radical scavenging activities than that with LO peel addition (*p* < 0.05). Antibacterial analysis of ABT synbiotic yoghurt with citrus peel addition showed that the diameters of inhibition zones against *S. aureus*, *B. subtilis*, and *E. coli* increased by 0.6–1.9 mm relative to the control groups, suggesting the enhancement of antibacterial activities by citrus peels. The viabilities of probiotic starter cultures (*L. acidophilus*, *S. thermophilus*, and *Bifidobacterial* sp.) were also enhanced by the incorporation of citrus peels in synbiotic yoghurt during cold storage. Hence, our results suggest that citrus peels, especially SO and SWO peels, could be recommended as a promising multifunctional additive for the development of probiotic and synbiotic yoghurt with enhanced antioxidant and antibacterial properties, as well as probiotic viability.

## 1. Introduction

The leading area of fruit production around the world is citrus cultivation, with its over-production having been transformed with technology into other food products, such as jams and juices. However, these citrus fruit industries produce close to 120 million tons per year of citrus waste worldwide [1,2]. The waste can be seeds, pulp residues, and citrus peels which can cause environmental pollution [3,4]. The millions of tons of waste are an economic and ecological problem. A promising solution is to recover the citrus waste. One of the highly valuable constituents is citrus peel, which has many applications in the food, cosmetics, and nutraceutical industries, and also in the production of biofuels and biodegradable materials [5,6]. Citrus peel has also been known to be a good source of phenolic compounds with antiaging, antitumor, antimicrobial, and anti-inflammatory properties [7,8]. Additionally, citrus peel contains about 20–30% of pectin in its dry weight, which is a prebiotic polysaccharide with a multitude of health benefits and thus widely used as a functional ingredient in the food industry.

The positive effects of adding citrus peel or by-product to multiple products such as brioches, meatballs, marmalade, and jam, have been shown by previous researchers [9,10,11,12,13]. Caggia et al. reported that the addition of debittered orange fiber resulted in final brioche products with increased moisture and protein, decreased fat content, and good textural properties [10]. In a study by Younis et al. [11], the firmness and chewability of jam were improved due to the addition of sweet lemon peel. In marmalade, the increase in the contents of phenolic compounds and their antioxidant capacity was due to the addition of sweet orange peel powder, however, this also caused a reduction in pH, titratable acidity, and anthocyanin content [12]. Orange peel could improve the physicochemical and nutritional profile of jams with a sensory acceptance similar to standard products [13]. Thus, the addition of citrus fruit residues in food products may improve the sensory, physicochemical, nutritional profile of foods and also contribute to the preservation of the environment.

One of the widely popular fermented dairy products is yoghurt with special nutritional, therapeutic, and sensory properties. Probiotic yoghurt is produced by *acidophilus-bifidus-thermophilus* (ABT)-type cultures which have *L. acidophilus*, *Bifidobacteria* sp., and *S. thermophilus* as the main fermenting organisms. In addition, *L. acidophilus*, *L. casei*, and *L. paracasei* have been studied extensively in the previous studies on probiotic yoghurt [14]. Espírito-Santo et al. [15] focused on improving the growth of probiotic bacteria by adding some prebiotic ingredients for the enrichment of yoghurt quality. Adding fruits, nuts, inulin, and grains as a source of fiber to milk is an exemplary method of prebiotic fortification [16,17,18,19].

Multiple attempts, including the addition of fruit peels, have been made to increase the biological efficiency of probiotics due to the presence of phenolic compounds and pectin in fruit peels. For example, as reported previously, fat and sugar-free probiotic yoghurt has been prepared with the combined addition of fruit peels (orange, passion fruit, and pineapple), which led to an increase in consumer acceptability, firmness, lactobacillus counts, and a decrease in syneresis [20]. Until now, no studies have been reported on the use of citrus peels for improving the functional properties of ABT-type fermented milk. Moreover, Khalid [21] reported the maximum concentration of the fraction of monoterpene hydrocarbons in sour orange peel samples, while Klimek-Szczykutowicz et al. [22] found that the major secondary metabolites in lemon are flavonoids, phenolic acids, coumarins, carboxylic acids, and vitamins.

Accordingly, this work aimed to investigate the multifarious effects of sour orange (SO), sweet orange (SWO), and lemon (LO) peels on the typical physiochemical, antioxidative, and antibacterial activities, as well as the probiotic viability of ABT-type synbiotic yoghurt during cold storage. The polyphenolic composition in citrus peels was analyzed by the high-performance liquid chromatography-diode array detector (HPLC-DAD) method.

## 2. Materials and Methods

### 2.1. Materials and Chemicals

Fresh cow milk was purchased from the dairy processing unit of the Faculty of Agriculture, Cairo University. Freeze-dried ABT-2 starter cultures containing *Lactobacillus acidophilus*, *Bifidobacterial* sp., and *Streptococcus thermophilus* were obtained from Chr. Hansen laboratories, Copenhagen, Denmark. The indicator bacterial strains *Staphylococcus* (*S.*) *aureus* NRRL-B-120, *Bacillus* (*B.*) *subtilis* NRRL-B-356, and *Escherichia* (*E.*) *coli* NRRL-B-409 were obtained from the Northern Regional Research Laboratory (NRRL), Peoria, USA. De Man Regosa Sharpe (MRS), M17, MacConkey, and plate-count milk agar were purchased from Difco Co., Lawrence, USA. Anaerobic sachets were obtained from Ineos Oxide Ltd., Hampshire, UK. The SO (*Citrus aurantium*), SWO (*Citrus sinensis*), and LO (*Citrus limon*) fruits were purchased from the local market.

L-cysteine hydrochloride and lithium chloride were obtained from El-Nasr Chem. Co. (Cairo, Egypt). Ascorbic acid, 2,2-diphenyl-1-picrylhydrazyl (DPPH), standards of catechin, naringin, rutin, kaempferol, myricetin, quercetin, resveratrol, rosemarinic acid, cinnamic acid, *o*-coumaric acid, ferulic acid, *p*-coumaric acid, caffeic acid, chlorogenic acid, benzoic acid, ellagic acid, syringic acid, vanillic acid, *p*-hydroxybenzoic acid, and gallic acid were purchased from Sigma-Aldrich (St. Louis, MO, USA). All other reagents and solvents used were of at least analytical grade.

### 2.2. Preparation of Different Citrus Peel Powder

Citrus peel powder was prepared based on a previous method described by Al-Bedrani et al. [16] with minor modifications. Briefly, the SO, SWO, and LO fruits were washed with tap water and immersed in sodium hypochlorite solution (100 mg/L) for 5 min, and then rewashed with tap water. The peels were removed manually with stainless steel knife and cut into small pieces, and dried at 40 °C for 24 h by using a drying oven. The dried peel was grinded to a powder form (mesh size 100:0.150 mm) and stored at −18 °C until further use.

### 2.3. Preparation of ABT Synbiotic Yoghurt

A preliminary experiment was conducted to select the suitable percentages of the different citrus peel powders. Cow raw milk (fat, 3.2%; protein, 3.3%; total solids (TS), 12.4%) was heated to 45 °C and then divided into nine equal portions (3 kg each). Each portion was added with different concentrations (0.5%, 1.0%, and 2.0%, *w*/*w*) of each citrus peel powder, mixed well, and heat treatment was continued until 90 °C for 10 min. Thereafter, the samples were cooled to 40 °C, and each portion was inoculated with 0.02% freeze-dried ABT-2 starter culture. The different treatments were dispensed into 150 mL polystyrene cups and incubated at 40 °C until the titratable acidity reached 0.85–0.90% of lactic acid.

### 2.4. Determination of Polyphenols

Polyphenols in citrus peel were determined by the method described by Bridi et al. [23] with some modifications. Briefly, high performance liquid chromatography (HPLC) analysis was carried out using an Agilent 1260 infinity series apparatus (Agilent Technologies, Santa Clara, CA, USA), equipped with a Quatemary pump and an Agilent diode array detector. The analytical column was a Kinetex EVO-C18 column (100 mm × 4.6 mm length, 5 µm particle size) with C18 guard column (Phenomenex, Torrance, CA, USA), operated at 30 °C. The separation was achieved using a ternary mobile phase of methanol (A), acetonitrile (B), and 0.2% H_3_PO_4_ (*v*/*v*) in HPLC grade water (C) at 0.7 mL/min. The gradient elution program was: 20% B/80% C 0–5 min, 7.5% A/25% B/67.5% C 5.1–10 min, 15% A/25% B/60% C 10.1–18 min, and 5% A/45% B/40% C 18.1–28 min. The injected volume was 20 µL. The peaks were monitored with a DAD set in the wavelength range of 200–650 nm. Phenolic acids and resveratrol were detected at 284 nm, and flavonoids were detected at 350 nm. The chromatograms were integrated for citrus peel samples at 284 nm. All samples were filtered through a 0.45 μm syringe filter before injection. The peaks were identified by comparing the retention time to the standards of polyphenols. All identified polyphenols were quantified by the external standard method via the respective standard curve which were obtained using the multistandard solution at concentrations ranging from 0.005 to 5.000 mg/L. The quantification limits of the polyphenols were determined to be in the range of 0.005–0.239 μg/L. The validation parameters of this method are listed in Appendix A. All analyses were conducted in triplicate. The phenolic acid levels were expressed as micrograms per gram of dry weight (μg/g DW).

For the identification of polyphenolic compounds extracted from citrus peels, high-performance liquid chromatography coupled with tandem mass spectrometry (HPLC-MS/MS) analysis was performed on an Agilent HPLC 1200 system (Agilent Technologies, Santa Clara, CA, USA) coupled with an Agilent triple-quadrupole mass spectrometer. The HPLC gradient system was the same as mentioned above. Samples were analyzed in negative mode (ESI-) and the injection volume was 20 μL. The ESI conditions were set as follows: capillary voltage of 0.8 kV, desolvation temperature of 600 °C, ion source temperature of 350 °C, and desolvation gas flow rate of 35 L/min. Qualification was based on multiple reaction monitoring (MRM) of selected ion pairs. The mass spectra were analyzed using Agilent MassHunter Qualitative Analysis software (Version B.06.00).

### 2.5. Sensory Evaluation

A panel of ten assessors was selected and trained as described in the ISO standard [17]. The Hedonic scale (1: dislike very much, 9: like very much) was used for evaluating the overall acceptability of the different ABT synbiotic yoghurt formulates.

### 2.6. Titratable Acidity (%) and Moisture Content

The titratable acidity and moisture content of each sample was measured according to the American Public Health Association (APHA) method [24].

### 2.7. 2,2-Diphenyl-1-picrylhydrazyl (DPPH) Free Radical Scavenging Activity

Crude extract of each treatment was prepared as previously described by Virtanen et al. [25]. Aliquots were collected from the different ABT synbiotic yoghurt treatments and the pH was adjusted to 4.6 and then the samples were centrifuged at 10,000× *g* for 15 min. The supernatant was filtered through a 0.45 µm sterilized filter. The antioxidant activity of the different ABT synbiotic yoghurt treatments was assessed based on the scavenging activity of DPPH free radicals which was calculated by using the following equation: Radical scavenging rate (%) = (*A*_blank_ − *A*_sample_)/*A*_blank_ × 100, where *A*_blank_ is the absorbance of the control and *A*_sample_ is the absorbance of the samples. Ascorbic acid was used as a positive control.

### 2.8. Antibacterial Activity of Different ABT Synbiotic Yoghurts

The antibacterial activity of the crude extract of the three different synbiotic yoghurt treatments was determined against *S. aureus*, *B. subtilis*, and *E. coli* strains using agar-well diffusion based on the previously reported method by Hassan et al. [26]. The bacterial cultures were incubated overnight at 37 °C, followed by dilution to yield a standardized inoculum of 1.5 × 10^8^ CFU/mL. A 0.5 mL of sterile crude extract of each treatment was transferred to each well at 0, 14, and 28 days of storage period. A paper disc with 100 μL of crude extract was placed on the surface of the agar plates, which were incubated at a 37 °C for 18 h. The diameters of the inhibition zones (mm) were measured by subtracting the disc diameter (mm) from the clear zone diameter (mm).

### 2.9. Viability of ABT Starter Culture in Different ABT Synbiotic Yoghurts during Cold Storage

The viability of the ABT starter culture in different synbiotic yoghurt treatments was assessed during the cold storage period. MRS (pH 5.5) agar was used for enumeration of *L. acidophilus* [4]. MRS agar supplemented with 0.05% (*w*/*v*) L-cysteine hydrochloride and 0.3% (*w*/*v*) lithium chloride was used for counting the viable cells of *Bifidobacteria* sp. [27]. However, M17 agar was used for enumerating the viable count of *S. thermophilus* [28]. Plates of *L. acidophilus* and *S. thermophilus* were incubated aerobically, however, plates of *Bifidobacteria* sp. were incubated anaerobically using an anaerobic jar (Ineos Oxide Ltd., Hampshire, UK).

### 2.10. Statistical Analysis

Data were analyzed using statistical methods by the Tukey test in the SPSS V11.5 for Windows (SPSS Inc., Chicago, IL, USA). The results were expressed as the mean ± standard deviations (SD) of triplicate independent experiments. Differences between means were considered significant at *p* < 0.05.

## 3. Results and Discussion

### 3.1. Determination of Polyphenols in Fruit Peels

The polyphenolic compounds were analyzed in SO, SWO, and LO peels by using HPLC. The chromatograms (Appendix A) integrated at 284 nm show that there were different chemical components, including three classes of polyphenols such phenolic acids, stilbenes, and flavonoids in the tested fruit peels (Table 1). The total amount of polyphenols in the SO peel (6.86 mg/g DW) was greatly higher than that in the SWO (3.19 mg/g DW) and LO peels (2.53 mg/g DW). The predominant phenolic compounds were myricetin (2.10 mg/g DW) and *o*-coumaric acid (1.13 mg/g DW) in the SO peel, benzoic acid (0.81 mg/g DW) and naringin (0.72 mg/g DW) in the SWO peel, benzoic acid (0.76 mg/g DW) and quercetin (0.36 mg/g DW) in the LO peel. It was also observed that a large amount of resveratrol was detected in the SO peel but not in the SWO peel. In addition, ferulic acid and rosemarinic acid were not detected in the SWO peel. The LO peel showed the lowest amount of all types of polyphenols, but *p*-coumaric acid and rutin were only detected in the LO peel. The results were consistent with the previous report by Marzouk [29]. In addition, caffeic acid, gallic acid, *p*-coumaric acid, and catechin compounds were found in high quantity in orange peel [30]. Furthermore, myricetin has been demonstrated to existed in fruits and vegetables, and its antibacterial, antiviral and antioxidant, anti-inflammatory, and anticancer activities have been evaluated [31]. Huang et al. [32] reported that hesperidin, naringin, neohesperidin, narirutin, and eriocitrin were the major flavonoids in eight species of citrus peel extracts. Gómez-Mejía et al. [33] identified ferulic acid, *p*-coumaric acid, naringin, and rutin in all tested citrus peel extracts, and suggested that citrus peel could be considered as a high source of polyphenols for value-added products.

### 3.2. Sensory Evaluation of ABT Synbiotic Yoghurt with Citrus Peel Addition

To understand the overall sensory acceptability of the ABT synbiotic yoghurt with citrus peel addition, sensory evaluation was conducted in the present work, and the mean scores for the hedonic scale of ABT synbiotic yoghurt with different concentrations of citrus peel addition (0.5–2.0%) are shown in Figure 1. It was observed that the addition of citrus peels dose-dependently affected the overall acceptability scores of the ABT synbiotic yoghurt. Regardless of the type of added citrus peel, the overall acceptability scores of the ABT synbiotic yoghurt with 0.5% citrus peel addition did not change significantly compared to the control without citrus peel addition (*p* > 0.05). However, the overall acceptability scores decreased significantly with increasing addition (1.0–2.0%) of SO, SWO, and LO peels into the ABT synbiotic yoghurt, suggesting that a high-concentration addition (>0.5%) of citrus peels negatively affected the overall acceptability of the ABT synbiotic yoghurt. The sequence of influence-degree of citrus peels for the overall sensory acceptability of the ABT synbiotic yoghurt was listed as LP peel > SO peel > SWO peel. Therefore, the ABT synbiotic yoghurt with 0.5% of selected citrus peels was further investigated for the cold storage test.

To our knowledge, sensory evaluation plays an important role in the development of a wide range of probiotic dairy food products [34,35]. Our results in Figure 1 show that fortification of the ABT synbiotic yoghurt with 0.5% of each peel powder had the highest overall acceptability scores by panelists. These results were in agreement with Dias et al. [20] who found that fortification of set yoghurt with 0.5% of composite fruit peel powder had the highest score of overall acceptability.

### 3.3. Titratable Acidity and Moisture Content of ABT Synbiotic Yoghurt with Citrus Peel Addition during Cold Storage

The dynamic variation in titratable acidity and moisture of the ABT synbiotic yoghurt with the addition of different citrus peels during cold storage (5 ± 1 °C) are represented in Table 2. The moisture content in all of the tested ABT synbiotic yoghurts with/without citrus peel addition did not change significantly during the cold storage period (*p* > 0.05). However, regardless of storage time and the type of citrus peel, the moisture contents in the ABT synbiotic yoghurts with citrus peel addition decreased slightly compared to that of the control. On the contrary, the titratable acidity of all of the tested ABT synbiotic yoghurts increased significantly throughout the storage period, especially for ABT synbiotic yoghurt fortified with different citrus peels (*p* < 0.05).In addition, the titratable acidities of the ABT synbiotic yoghurts fortified with citrus peels were significantly higher than that of the control for the same storage time (14 and 28 days). The results were in agreement with a previous report which showed that the addition of different ratios of orange marmalade to yoghurt decreased the pH and increased the acidic flavor [16]. It was also reported that yoghurt incorporated with orange fiber as the main component of citrus peel had a significant increase in acidity compared to the control [17]. However, no difference was observed among the ABT synbiotic yoghurts with SO, SWO, and LO peel addition, which indicated that the addition of 0.5% citrus peels did not lead to a change in the titratable acidities of the ABT synbiotic yoghurts compared to control. As shown in Appendix A, the total polyphenol contents in the ABT synbiotic yoghurts fortified with SO, SWO, and LO were 32.94, 15.00, and 11.40 µg/g, respectively. The obvious difference in the total polyphenol contents among the synbiotic yoghurts fortified by different citrus peels was also inconsistent with the same titratable acidities among them during storage for 14 and 28 days. Accordingly, these observations suggest that the increase in titratable acidity in the synbiotic yoghurts with citrus peel addition had little relation to the polyphenols in citrus peels. With regard to moisture content, the values of the moisture content significantly decreased when different citrus peel powders were added to the ABT synbiotic yoghurt in comparison with the control. The results obtained by Pastorino et al. [36] illustrated that an increase in the acidity resulted in cheese with a low moisture content.

### 3.4. Antioxidant Activity of ABT Synbiotic Yoghurt with Citrus Peel Addition

The antioxidant activities of the ABT synbiotic yoghurts with different citrus peel additions were measured by a DPPH free radical scavenging assay, and the results are shown in Figure 2. As expected, the DPPH radical scavenging activities of the ABT synbiotic yoghurts with citrus peel addition were significantly higher than that of the control without citrus peel addition, which was attributed to the presence of polyphenolic compounds in added citrus peels. Due to the relative higher contents of polyphenolic compounds in SO and SWO peels than that in LO peel (Table 1), which was also observed in the ABT synbiotic yoghurt fortified by citrus peels (Appendix A), the ABT synbiotic yoghurts fortified with 0.5% of SO (80.55%) and SWO (79.15%) had a higher antioxidant capacity than the LO-fortified ABT synbiotic yoghurt (71.10%) (*p* < 0.05). Additionally, the DPPH radical scavenging activity of the synbiotic yoghurt without citrus peel addition increased significantly (*p* < 0.05) compared to that of unfermented milk, suggesting that some antioxidant substances could be generated during milk fermentation. As reported previously, antioxidant peptides were produced from α-lactalbumin, β-lactoglobulin, and α-casein during milk fermentation [37,38], which mainly contributed to enhanced antioxidant activity in fermented dairy products compared to that in unfermented milk [39]. However, Moschopoulou et al. reported that no significant proteolysis was detected in set-type yoghurt during storage [40]. Therefore, the above observations indicate that the polyphenolic composition of citrus peels contributed greatly to the enhancement of the antioxidant activity of the novel ABT synbiotic yoghurt formulates during storage. Huang et al. [32] evaluated the antioxidant activities of eight species of citrus peel extracts and reported that ponkan peel extract had the greatest overall antioxidant activity. However, Czech et al. [41] found that the pulp of oranges and all grapefruit varieties could scavenge the DPPH radicals to a significantly higher extent than in the peel. Recently, fortification of yoghurt drinks with different citrus peel powders has enhanced their antioxidant capacity during the shelf life of the product [42].

### 3.5. Antibacterial Activity of ABT Synbiotic Yoghurt with Citrus Peel Addition during Cold Storage

The antibacterial activities of different probiotic strains play an important role as bio-preservatives in a wide range of dairy foods [43,44]. In the present work, the antibacterial activity of the ABT synbiotic yoghurt was evaluated by a disc diffusion assay, which was expressed as the inhibition zone against pathogenic strains. From Table 3, the inhibition zone of the crude extracts from ABT milk with/without citrus peel addition increased with prolonged storage time. The increasing antibacterial activity of the ABT milk during cold storage might be attributed to the increasing acidity of the ABT milk, which could inhibit the growth of pathogenic strains [45]. Additionally, the inhibition zone of the ABT milk with citrus peel addition increased obviously compared to the control in the same storage time. Taking the ABT milk with SO peel addition as example, we found that the inhibition zone increased from 6.50 to 7.40 mm at day 0, 7.00 to 8.20 mm at day 14, and 7.80 to 9.30 mm at day 28. Thus, fortification of the ABT synbiotic yoghurt with 0.5% of different peel powders significantly enhanced the antibacterial activity against *S. aureus*, *B. subtilis*, and *E. coli* compared to the control for the same storage period (Table 3). From the general comparison of the inhibition zone for these three pathogenic strains, we observed that the antibacterial activities of citrus peels against gram-positive bacteria (*S. aureus* and *B. subtilis*) was stronger than that against gram-negative bacteria (*E. coli*) in the ABT synbiotic yoghurt. The results were consistent with a previous report in which it was well confirmed that citrus peel extract including high contents of polyphenols had a strong antibacterial activity [46]. The enhancement of antibacterial efficiency might be attributed to the polyphenolic compounds in citrus peel.

Furthermore, probiotics in synbiotic yoghurt could also contribute to the enhancement of antibacterial activities. As reported previously, *Lactobacilli* and *Bifdobacterium strains* had antibacterial activities against *S. aureus*, *B. subtilis*, *Enterococcus aerogenes*, and *Ps. fluorescence* [47]. In another research, results indicated that *L. bulgaricus* with *S. thermophilus* had the highest antibacterial activity against *S. aureus* with an inhibition zone of 10.5 mm and for *E. coli* with 4.0 mm [48]. Different *Lactobacillus* spp. possess varied inhibitory activity against *E. coli*, *S. aureus*, *B. cereus*, *B. subtilis*, and *S. typhi* [48,49,50,51]. Fortification of yoghurt drinks with orange and lemon peel powders enhanced the antibacterial and antifungal activities of the products during shelf-life period [42].

### 3.6. Viability of ABT Starter Culture of Synbiotic Yoghurt with Citrus Peel Addition during Cold Storage

The viability of three mixed ABT starter cultures containing *L. acidophilus*, *S. thermophilus*, and *Bifidobacteria* sp. was assessed for synbiotic yoghurt with citrus peel addition during cold storage, and the results are shown in Table 4. Generally, the viability of *L. acidophilus*, *Bifidobacterial* sp., and *S. thermophilus* in the ABT synbiotic yoghurt without citrus peel addition decreased significantly during cold storage (*p* < 0.05), which might be because the cold storage was represented as physiological stress on the viability of the ABT starter cultures. However, almost no decrease in the viability of these three probiotics was observed in the ABT milk with 0.5% of different citrus peels. This finding indicates that fortification of the ABT synbiotic yoghurt with citrus peel addition enhanced the viability of *L. acidophilus*, *S. thermophilus*, and *Bifidobacteria* sp. during cold storage. The findings may be attributed to the fiber content in citrus peels which acts as a growth promoter (prebiotic effect) for probiotics. As reported previously, the addition of fruit fiber increased the numbers of *S. thermophilous* and *L. bulgaricus* [20], and fortification of yoghurt with pineapple peel could enhance the viability of *L. acidophilus* as well as *L. paracasei* ssp. *paracasei* during the cold storage period [52]. It has also been confirmed that citrus pectin hydrolysate can enhance the growth of probiotics including *L. acidophilus* and *B. bifidum* [53]. Exceptionally, the viabilities of *S. thermophilus*, and *Bifidobacteria* sp. were not affected by the addition of citrus peels in the ABT milk at 14 days of storage, but significantly increased at 28 days of storage (*p* < 0.05). The enhancement in the viability of the ABT starter culture thus led to an increase in the titratable acidities of the ABT synbiotic yoghurts with citrus peel addition (Table 2). These above results matched with the report by Erkaya-Kotan [17] who reported that in the first 7 days of storage, the viable counts of *S. thermophilus* in yoghurt samples were decreased compared to the control, and then slightly increased. Casarotti et al. [54] evaluated the addition of fruit by-product to fermented products and found that the population of *S. thermophiles* remained stable during the storage period. The decrease or invariability in the viable count of *S. thermophilus* was due to the inhibiting effect of lactic acid [55,56].

## 4. Conclusions

The addition up to 0.5% of different types of citrus (SO, SWO, and LO) peel powders in milk did not change the overall acceptability scores of the ABT synbiotic yoghurt statistically significantly (*p* < 0.05). The total phenolic content in the SO peel was more than 2-fold higher than that in the SWO and LO peel. Vanillic and syringic acids were found in the SO and SWO peels, but not in the LO peel. *p*-Coumaric acid and rutin were only found in the LO peel. The addition of citrus peels led to an increased acidity and decreased moisture of the ABT synbiotic yoghurt during cold storage (14 and 28 days). Furthermore, the supplementation with citrus peels also improved the antioxidant and antibacterial activities of the ABT synbiotic yoghurt. The ABT milks with SO and SWO peel addition had significantly stronger DPPH radical scavenging activities than that with LO peel addition. Additionally, the viabilities of the probiotic starter cultures were also enhanced by the incorporation of citrus peels in synbiotic yoghurt during cold storage. Therefore, this work provides valuable information about the promising potential of citrus peels, particularly SO and SWO peels, as multifunctional food additives applied in ABT-type synbiotic yoghurt.

## Figures and Tables

**Figure 1 foods-11-02677-f001:**
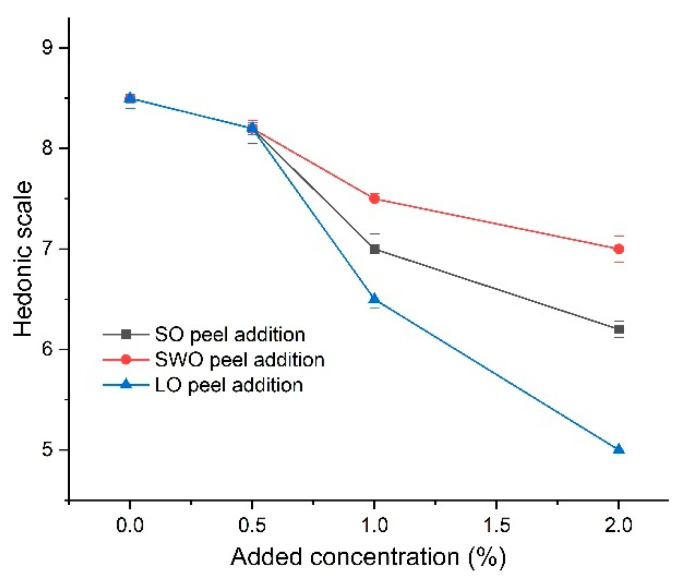
Overall acceptability (9-point hedonic scale) of ABT synbiotic yoghurt with different citrus peel addition. SO: sour orange peel powder, SWO: sweet orange peel, LO: lemon peel powder.

**Figure 2 foods-11-02677-f002:**
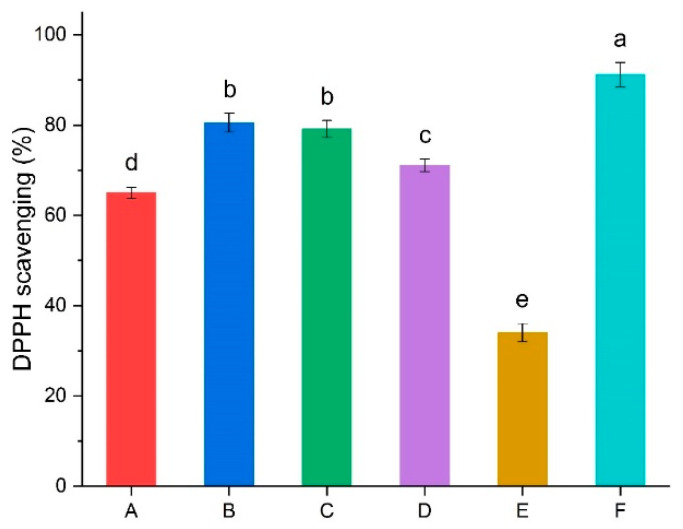
Antioxidant activity of ABT synbiotic yoghurt with different citrus peel addition. (A): ABT synbiotic yoghurt without citrus peel addition (control), (B): ABT synbiotic yoghurt with 0.5% SO peel addition, (C): ABT synbiotic yoghurt with 0.5% SWO peel addition, (D): ABT synbiotic yoghurt with 0.5% LO peel addition, (E): unfermented milk, (F): ascorbic acid. Different letters in the same column indicate a significant difference between the treatments at *p* < 0.05.

**Table 1 foods-11-02677-t001:** Profiles of polyphenolic compounds (µg/g DW) in different citrus peels.

	Compounds	SO Peel	SWO Peel	LO Peel
Phenolic acids				
1	*o*-Coumaric acid	1128.23 ± 23.48 ^b^	266.22 ± 10.61 ^e^	14.13 ± 0.96 ^k^
2	Benzoic acid	972.78 ± 9.72 ^c^	809.58 ± 13.67 ^a^	758.68 ± 20.11 ^a^
3	Ellagic acid	604.76 ± 10.43 ^e^	249.29 ± 9.46 ^e^	68.19 ± 3.76 ^g^
4	*p*-Hydroxybenzoic acid	92.26 ± 1.56 ^g^	194.20 ± 11.48 ^f^	281.43 ± 6.82 ^c^
5	Chlorogenic acid	61.72 ± 1.52 ^h^	35.09 ± 2.43 ^hi^	127.29 ± 12.88 ^f^
6	Caffeic acid	25.86 ± 2.09 ^I^	35.21 ± 2.85 ^hi^	34.05 ± 2.46 ^h^
7	Cinnamic acid	11.56 ± 1.04 ^k^	8.04 ± 1.18 ^k^	3.76 ± 0.58 ^l^
8	Gallic acid	6.05 ± 0.95 ^l^	6.52 ± 0.86 ^k^	19.47 ± 1.82 ^j^
9	Vanillic acid	23.95 ± 2.08 ^i^	59.43 ± 1.39 ^g^	n.d.
10	Syringic acid	17.43 ± 0.91 ^j^	31.13 ± 3.09 ^i^	n.d.
11	Ferulic acid	18.53 ± 1.76 ^j^	n.d.	36.28 ± 1.08 ^h^
12	Rosemarinic acid	12.63 ± 1.27 ^k^	n.d.	73.26 ± 2.96 ^g^
13	*p*-Coumaric acid	n.d.	n.d.	24.82 ± 1.49 ^i^
Stilbenes				
1	Resveratrol	755.94 ± 8.93 ^d^	n.d.	77.57 ± 6.45 ^g^
Flavonoids				
1	Myricetin	2098.10 ± 38.12 ^a^	345.79 ± 15.39 ^d^	153.39 ± 5.91 ^e^
2	Quercetin	615.68 ± 11.38 ^e^	379.01 ± 10.85 ^c^	364.15 ± 15.27 ^b^
3	Naringin	381.03 ± 10.82 ^f^	719.94 ± 14.22 ^b^	269.87 ± 3.99 ^c^
4	Kaempferol	6.68 ± 1.04 ^l^	16.24 ± 1.64 ^j^	20.57 ± 2.63 ^j^
5	Catechin	22.57 ± 1.83 ^i^	38.97 ± 2.85 ^h^	26.08 ± 1.75 ^i^
6	Rutin	n.d.	n.d.	181.10 ± 5.24 ^d^
Total		6855.77	3194.67	2534.09

Data are presented as means ± SD. Different letters in the same column indicate a significant difference at *p* < 0.05. SO: sour orange, SWO: sweet orange, LO: lemon. n.d.: not detected.

**Table 2 foods-11-02677-t002:** Titratable acidity and moisture of ABT synbiotic yoghurt with citrus peel addition during cold storage (5 ± 1 °C).

Experimental ABT Synbiotic Yoghurt	Storage Period (Day)	Titratable Acidity (% Lactic Acid)	Moisture (%)
A: Control (without citrus peel addition)	0	0.87 ± 0.05 ^c^	87.54 ± 0.15 ^a^
	14	0.92 ± 0.04 ^c^	87.48 ± 0.19 ^a^
	28	1.15 ± 0.06 ^b^	87.50 ± 0.14 ^a^
B: 0.5% SO peel addition	0	0.90 ± 0.03 ^c^	86.90 ± 0.21 ^b^
	14	1.20 ± 0.04 ^ab^	86.85 ± 0.13 ^b^
	28	1.30 ± 0.06 ^a^	86.78 ± 0.20 ^b^
C: 0.5% SWO peel addition	0	0.90 ± 0.04 ^c^	86.92 ± 0.14 ^b^
	14	1.15 ± 0.07 ^b^	86.80 ± 0.09 ^b^
	28	1.23 ± 0.03 ^ab^	86.75 ± 0.07 ^b^
D: 0.5% LO peel addition	0	0.92 ± 0.03 ^c^	86.88 ± 0.16 ^b^
	14	1.20 ± 0.02 ^b^	86.80 ± 0.11 ^b^
	28	1.23 ± 0.06 ^ab^	86.78 ± 0.06 ^b^

Values are expressed as the means ± standard deviation (n = 3). Different letters in the same column indicate a significant difference at *p* < 0.05.

**Table 3 foods-11-02677-t003:** Antibacterial efficiency of crude extract from ABT synbiotic yoghurt with citrus peel addition during cold storage (5 ± 1 °C).

Experimental ABT Synbiotic Yoghurt	Storage Period (Day)	Inhibition Zone (mm)		
		*S. aureus*	*B. subtilis*	*E. coli*
A: Control (without citrus peel addition)	0	6.50 ± 0.40 ^d^	5.10 ± 0.60 ^e^	3.70 ± 0.50 ^e^
	14	7.00 ± 0.60 ^d^	6.00 ± 0.50 ^de^	4.30 ± 0.20 ^de^
	28	7.80 ± 0.30 ^bc^	6.90 ± 0.40 ^cd^	6.30 ± 0.30 ^b^
B: 0.5% SO peel addition	0	7.40 ± 0.60 ^bcd^	6.20 ± 0.50 ^de^	4.40 ± 0.20 ^de^
	14	8.20 ± 0.50 ^bc^	7.30 ± 0.10 ^c^	6.30 ± 0.40 ^abc^
	28	9.30 ± 0.40 ^a^	8.20 ± 0.20 ^a^	7.20 ± 0.50 ^a^
C: 0.5% SWO peel addition	0	7.50 ± 0.50 ^bcd^	6.10 ± 0.30 ^d^	4.60 ± 0.20 ^d^
	14	8.20 ± 0.40 ^bc^	6.85 ± 0.60 ^cd^	5.80 ± 0.10 ^c^
	28	9.50 ± 0.20 ^a^	7.50 ± 0.50 ^abc^	6.50 ± 0.20 ^ab^
D: 0.5% LO peel addition	0	7.80 ± 0.10 ^c^	6.25 ± 0.20 ^d^	4.55 ± 0.10 ^d^
	14	8.30 ± 0.30 ^b^	7.10 ± 0.30 ^c^	5.40 ± 0.40 ^c^
	28	9.70 ± 0.20 ^a^	7.80 ± 0.10 ^b^	6.10 ± 0.50 ^bc^

Values are expressed as the means ± standard deviation (n = 3). Different letters in the same column indicate a significant difference at *p* < 0.05.

**Table 4 foods-11-02677-t004:** Viable counts (Log CFU/g) of *L. acidophilus*, *S. thermophilus*, and *Bifidobacteria* sp. in ABT synbiotic yoghurt with citrus peel addition during cold storage (5 ± 1 °C).

Experimental ABT Synbiotic Yoghurt	Storage Period (Day)	*L. acidophilus*	*S. thermophilus*	*Bifidobacteria* sp.
A: Control (without citrus peel addition)	0	8.30 ± 0.65 ^a^	8.45 ± 0.70 ^a^	8.05 ± 0.72 ^ab^
	14	6.90 ± 0.57 ^b^	7.75 ± 0.61 ^ab^	7.20 ± 0.83 ^abc^
	28	5.20 ± 0.62 ^c^	6.05 ± 0.85 ^b^	5.45 ± 1.10 ^c^
B: 0.5% SO peel addition	0	8.25 ± 1.20 ^ab^	8.35 ± 1.05 ^ab^	8.20 ± 0.81 ^a^
	14	7.50 ± 0.95 ^ab^	7.30 ± 0.85 ^ab^	7.55 ± 0.67 ^ab^
	28	6.10 ± 0.82 ^bc^	6.70 ± 0.71 ^b^	6.80 ± 0.57 ^bc^
C: 0.5% SWO peel addition	0	8.30 ± 0.61 ^a^	8.20 ± 0.63 ^a^	8.30 ± 1.06 ^ab^
	14	7.20 ± 0.55 ^ab^	7.62 ± 0.75 ^ab^	7.10 ± 1.01 ^abc^
	28	6.30 ± 0.68 ^bc^	6.85 ± 0.61 ^b^	6.70 ± 0.82 ^abc^
D: 0.5% LO peel addition	0	8.35 ± 0.81 ^a^	8.40 ± 0.71 ^a^	8.25 ± 0.92 ^ab^
	14	7.10 ± 1.10 ^ab^	7.75 ± 0.86 ^ab^	7.20 ± 0.61 ^ab^
	28	6.15 ± 1.22 ^bc^	6.52 ± 0.91 ^b^	6.65 ± 0.80 ^abc^

Values are expressed as the means ± standard deviation (n = 3). Different letters in the same column indicate a significant difference at *p* < 0.05.

## Data Availability

Data is contained within the article.

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
