# Peer review of "Value-Added Utilization of Citrus Peels in Improving Functional Properties and Probiotic Viability of Acidophilus-bifidus-thermophilus (ABT)-Type Synbiotic Yoghurt during Cold Storage"

_foods, 2022, doi:10.3390/foods11172677_

Round 1
Reviewer 1 Report
The paper aims to enhance the functionality and probiotic growth by adding citrus peels in milk and subsequently, yoghurt. However, in my opinion, if prebiotic substances are also incorporated to enhance the probiotic viability of the organisms, the resultant product should be named as ‘synbiotic yoghurt’. The title can be modified accordingly.
1. Introduction
Line 41: Add reference for the amount of waste production.
Line 44: What do you mean by ‘mentioned size of residues’?
Line 53: ‘Shown by previous researchers’. Cite few of the research at the end of this line for authenticating a scientific sentence.
https://doi.org/10.3390/ijms12042174
https://doi.org/10.1016/j.foodres.2018.01.075
https://doi.org/10.3390/molecules25071670
https://doi.org/10.3390/molecules25122812
https://doi.org/10.3390/foods11142030
https://doi.org/10.3390/foods10112855
Line 68: More research studies must be added here, which have focused on enhancing the viability of probiotic organisms by adding one or the other prebiotic substances. Polysaccharides such as inulin also acts as prebiotic organisms.
https://doi.org/10.1111/jfpp.15322
https://doi.org/10.3390/app10124317
Introduction basically lacks specific aim and investigation. Kindly fill the gap and re-write the last paragraph of the introduction.
2. Materials and Methods
Line 113: No mention of batch size in the methodology.
Line 165: Equation for the standard curve of DPPH is not given.
3. Results and Discussion
Line 238: It is always better to present sensory scores in the form of spider-web diagram.
Line 250-259: Results of pH, acidity etc should be correlated to HPLC profile of citrus peel. Just comparing with previous study won’t give any novelty to this paper.
4. Conclusion
Conclusion should highlight the exact percent increase of polyphenols in various citrus peels used for this study. Merely, repeating the results is not the aim of writing the conclusion.
Author Response
1. The paper aims to enhance the functionality and probiotic growth by adding citrus peels in milk and subsequently, yoghurt. However, in my opinion, if prebiotic substances are also incorporated to enhance the probiotic viability of the organisms, the resultant product should be named as ‘synbiotic yoghurt’. The title can be modified accordingly.
Response: Thank you for your good suggestion. We have revised the title in the revised manuscript.
2. Line 41: Add reference for the amount of waste production.
Response: Thank you for your good suggestion. Due to our carelessness, the data of annual waste production amount was written wrongly. We have revised it and also added the reference for the amount of waste production.
3. Line 44: What do you mean by ‘mentioned size of residues’?
Response: Thank you for your question. We are sorry for the unclear expression. Actually, “mentioned size of residues” referred to the annual citrus waste mentioned above. To eliminate the misunderstanding, we have revised the expression as follows in the revised manuscript.
Original sentence: A promising solution is to recover mentioned size of residues.
Revised sentence: A promising solution is to recover the citrus waste.
4. Line 53: ‘Shown by previous researchers’. Cite few of the research at the end of this line for authenticating a scientific sentence.
Response: Thank you for your good suggestion. We have added few of references at the end of line 53 in the revised manuscript.
5. Line 68: More research studies must be added here, which have focused on enhancing the viability of probiotic organisms by adding one or the other prebiotic substances. Polysaccharides such as inulin also acts as prebiotic organisms.
https://doi.org/10.1111/jfpp.15322
https://doi.org/10.3390/app10124317
Response: Thank you for your good suggestion. We have added more references kindly recommended by you in line 76 in the revised manuscript.
6. Introduction basically lacks specific aim and investigation. Kindly fill the gap and re-write the last paragraph of the introduction.
Response: Thank you for your good suggestion. We have rewritten the last paragraph of the introduction in the revised manuscript.
7. Line 113: No mention of batch size in the methodology.
Response: Thank you for your good comment. We have added the batch size (27 kg per batch) in section 2.3 in the revised manuscript.
8. Line 165: Equation for the standard curve of DPPH is not given.
Response: Thank you for your good comment. We have added the equation of DPPH radical scavenging rate in the revised manuscript.
9. Line 238: It is always better to present sensory scores in the form of spider-web diagram.
Response: Thank you for your comment. The 9-point hedonic scale is the most widely used scale for measuring food acceptability, which can well assess the average degree of liking or disliking of foods or consumer products at an overall level. Thus, in our work, it was used to evaluate whether ABT fermented milk fortified by citrus peel was overall accepted or not. Furthermore, Figure 1 just showed overall acceptability of ABT synbiotic yogurt with different citrus peel addition, and thus maybe it is difficult to be shown in the form of spider-web diagram.
10. Line 250-259: Results of pH, acidity etc should be correlated to HPLC profile of citrus peel. Just comparing with previous study won’t give any novelty to this paper.
Response: Thank you for your comment. We have done the discussion about the relation between the acidities and polyphenol contents in synbiotic yogurt fortified by citrus peels. From Table 2, no difference was observed among ABT synbiotic yogurts with SO, SWO, and LO peel ad-dition, which indicated that the addition of 0.5% citrus peels did not lead to the change of titratable acidities of ABT synbiotic yogurts compared to control. As shown in Table S2, total polyphenol contents in ABT synbiotic yogurts fortified with SO, SWO, and LO were 32.94, 15.00, and 11.40 µg/g, respectively. The obvious difference in total polyphenol con-tents among synbiotic yogurts fortified by different citrus peels was also inconsistent with the same titratable acidities among them during storage for 14 and 28 days. Accordingly, these observations suggested that the increase of titratable acidity in synbiotic yogurts with citrus peel addition had little relation to polyphenols in citrus peels.
11. Conclusion should highlight the exact percent increase of polyphenols in various citrus peels used for this study. Merely, repeating the results is not the aim of writing the conclusion.
Response: Thank you for your good suggestion. According to your suggestion, we have revised the conclusion in the revised manuscript.
Reviewer 2 Report
Well written and designed article. Congrats to authors.
Here are my Minor suggestions for corrections.
All are listed below with an appropriate Line number(s) from text in order to facilitate tracking:
Line 28: Define also here that antioxidant properties were improved compared to control sample.
Line 30: I think you should abbreviate this "Staph." just as "S."? Like you do with B and E for the next microbes? If it is so that this issue should be corrected through a whole text.
Line 47: typo - delete surplus "." after "materials" and in front of the brackets.
Line 61: Delete this "in the world". It is surplus here.
Line 67: I think it should be here "dairy/yogurth" quality not "milk quality"? Please check/correct.
Line 80: "amino acids" are not secondary metabolites? Please check/correct.
Line 109: Please do not use ppm as unit for concentration. It is not acceptable in SI system. Correct adequately.
Line 162: typo - letter "g" after number should be given in Italic style. Correct.
Line 164: Please define here how you expressed your results for DPPH assay.
Line 261: typo - delete "," after "Pastorino". It is surplus here.
Line 277-279: I think that authors should fulfill discussion with details about possible additional antioxidants after fermentation process. Are there maybe some peptides obtained from proteins fermentation?
Line 284: The same comment as in the Line 261.
Lines 293-295: Question for authors- did you use some antibiotic as positive control during antibacterial assay? Please define/clarify.
Kind regards.
Author Response
1. Line 28: Define also here that antioxidant properties were improved compared to control sample.
Response: Thank you for your good suggestion. We had added the information about the antioxidant property of control sample in the revised manuscript.
2. Line 30: I think you should abbreviate this "Staph." just as "S."? Like you do with B and E for the next microbes? If it is so that this issue should be corrected through a whole text.
Response: Thank you for your careful review. According to your suggestion, we have abbreviated "Staph." as "S." in the whole manuscript.
3. Line 47: typo - delete surplus "." after "materials" and in front of the brackets.
Response: Thank you for your careful review. We have deleted surplus "." in line 47 in the revised manuscript.
4. Line 61: Delete this "in the world". It is surplus here.
Response: Thank you for your good suggestion. We have deleted “in the world” in the revised manuscript.
5. Line 67: I think it should be here "dairy/yogurth" quality not "milk quality"? Please check/correct.
Response: Thank you for your good comment. Agreeing with your suggestion, we have replaced "milk quality" with “yogurt quality” in the revised manuscript.
6. Line 80: "amino acids" are not secondary metabolites? Please check/correct.
Response: Thank you for your careful review. Indeed, amino acids are not secondary metabolites, but primary metabolites. We have revised this expression in the revised manuscript.
7. Line 109: Please do not use ppm as unit for concentration. It is not acceptable in SI system. Correct adequately.
Response: Thank you for your good suggestion. We have replaced “ppm” with “mg/L” in the revised manuscript.
8. Line 162: typo - letter "g" after number should be given in Italic style. Correct.
Response: Thank you for your careful review. We have revised it in the revised manuscript.
9. Line 164: Please define here how you expressed your results for DPPH assay.
Response: Thank you for your good suggestion. The scavenging activity of DPPH free radical was expressed as percentage (%) reduction of the initial DPPH absorbance compared to control, which has been added in the revised manuscript.
10. Line 261: typo - delete "," after "Pastorino". It is surplus here.
Response: Thank you for your careful review. We have deleted “,” after “Pastorino” in the revised manuscript.
11. Line 277-279: I think that authors should fulfill discussion with details about possible additional antioxidants after fermentation process. Are there maybe some peptides obtained from proteins fermentation?
Response: Thank you for your good suggestion. The details about possible additional antioxidants after fermentation process have been replenished in the revised manuscript. According to the previous reports, some antioxidant peptides were indeed produced from milk proteins during fermentation, which mainly led to the improvement of antioxidant activity of fermented dairy products compared to that of non-fermented milk (1. Li, S.; Tang, S.; He, Q.; Hu, J.; Zheng, J. In vitro antioxidant and angiotensin-converting enzyme inhibitory activity of fermented milk with different culture combinations. J. Dairy Sci. 2020, 103, 1120-1130. https://doi.org/10.3168/jds.2019-17165; 2. Tonolo, F.; Fiorese, F.; Moretto, L.; Folda, A.; Scalcon, V.; Grinzato, A.; Ferro, S.; Arrigoni, G.; Bindoli, A.; Feller, E. Identification of new peptides from fermented milk showing antioxidant properties: Mechanism of action. Antioxidants 2020, 9, 117. https://doi.org/10.3390/antiox9020117; 3. Khan, I.T.; Nadeem, M.; Imran, M.; Ullah, R.; Ajmal, M.; Jaspal, M.H. Antioxidant proper-ties of Milk and dairy products: A comprehensive review of the current knowledge. Lipids Health Dis. 2019, 18, 1-13. https://doi.org/10.1186/s12944-019-0969-8.)
12. Line 284: The same comment as in the Line 261.
Response: Thank you for your good comment. We have revised it in the revised manuscript.
13. Lines 293-295: Question for authors- did you use some antibiotic as positive control during antibacterial assay? Please define/clarify.
Response: Thank you for your question. In antibacterial assay, this work aimed to investigate antibacterial efficiency of crude extract from ABT fermented milk with different citrus peel addition compared to control during cold storage. Furthermore, it is not allowed to add antibiotic in fermented milk due to the presence of probiotics. Therefore, we did not used some antibiotic as positive control in antibacterial assay.
Reviewer 3 Report
1. The abstract should be revised, in case, conclusions and recommendations should of the study are present at the end of the abstract
2. line number 41, this data from which year and where?
3. section 2.2, the processing method is taken from some other studies or your own developed method?
4. line 111, what is the particle size?
5. In section 2.3 heading replace manufacturing with " preparation"
6. In section 2.4, give the reference for the method.
7. Section 2.5, why only considered overall acceptability? Other parameters are not taken?
8. I suggest to prepare a comparison or literature table of the phenolic acids reported by different author.
9. Line 257, Orange peel powder and fiber are different, how compared two?
10. Line 249, why moisture content was decreased?
11. Line 313, better to give the clear discussion on the antimicrobial activity issues
12. Add p-vales in the tables
13. line 214, replace “data are presented” as “data presented”
Author Response
1. The abstract should be revised, in case, conclusions and recommendations should of the study are present at the end of the abstract
Response: Thank you for your good suggestion. We have revised the abstract which includes the conclusions and recommendations of the study in the revised manuscript.
2. line number 41, this data from which year and where?
Response: Thank you for your good comment. We are sorry for the inaccurate data. Actually, from the previous reports (1. Kandemir, K.; Piskin, E.; Xiao, J.; Tomas, M.; Capanoglu, E. Fruit Juice Industry Wastes as a Source of Bioactives. J. Agric. Food Chem. 2022, 70, 6805-6832, https://doi.org/10.1021/acs.jafc.2c00756. 2. Russo, C.; Maugeri, A.; Lombardo, G.E.; Musumeci, L.; Barreca, D.; Rapisarda, A.; Cirmi, S.; Navarra, M. The Second Life of Citrus Fruit Waste: A Valuable Source of Bioactive Compounds. Molecules 2021, 26, https://doi.org/10.3390/molecules26195991.), about 120 million tons per year of citrus waste is produced worldwide. We have revised this sentence in line 41 in the revised manuscript.
3. section 2.2, the processing method is taken from some other studies or your own developed method?
Response: Thank you for your question. Actually, citrus peel powder was prepared based on a previous method with some minor modifications. We have added the reference in the revised manuscript.
4. line 111, what is the particle size?
Response: Thank you for your good question. Actually, the milled citrus peel powder passed through through a US mesh size 100 sieve (0.150 mm). Therefore, the particle size of prepared citrus peel powder is below 0.150 mm. We have replenished the information on the particle size of powder in the revised manuscript.
5. In section 2.3 heading replace manufacturing with " preparation"
Response: Thank you for your good suggestion. We have replaced “manufacturing” with “preparation” in the revised manuscript.
6. In section 2.4, give the reference for the method.
Response: Thank you for your good suggestion. We have added the reference for the method of polyphenol determination in the revised manuscript.
7. Section 2.5, why only considered overall acceptability? Other parameters are not taken?
Response: Thank you for your question. The 9-point hedonic scale is the most widely used scale for measuring food acceptability, which can well assess the average degree of liking or disliking of foods or consumer products at an overall level. Thus, in our work, it was used to evaluate whether ABT fermented milk fortified by citrus peel was overall accepted or not. Furthermore, the main purpose of our work is actually to investigate the effects of various citrus peels on typical physiochemical, antioxidative, antibacterial properties, and probiotic viability of ABT-type fermented milk during cold storage.
8. I suggest to prepare a comparison or literature table of the phenolic acids reported by different author.
Response: Thank you for your suggestion. The discussion of phenolic acids compared to the previous reports by different authors were presented in lines 215-223 in the revised manuscript. It was actually found that there was no big difference in polyphenol contents among our work and previous reports. Furthermore, the main purpose of our work is to investigate the effects of various citrus peels on typical physiochemical, antioxidative, antibacterial properties, and probiotic viability of ABT-type fermented milk during cold storage. Thus, the detailed comparison of phenolic acids was not shown in our manuscript. Now we are preparing a review paper focusing on citrus peel as food supplement, in which phenolic acids in citrus peels will be discussed in detail according to your valuable suggestion.
9. Line 257, Orange peel powder and fiber are different, how compared two?
Response: Thank you for your question. Actually, orange peel powder is a potent source of fiber including pectin, lignin, cellulose, and hemicellulose, and 1 g of orange peel powder contains ca 0.6 g of fiber (Marín, F.R.; Soler-Rivas, C.; Benavente-García, O.; Castillo, J.; Pérez-Alvarez, J.A. By-products from different citrus processes as a source of customized functional fibres. Food Chem. 2007, 100, 736-741, https://doi.org/10.1016/j.foodchem.2005.04.040.). As described in line 257, dietary fiber in citrus peel can lead to significant increase in acidity of yogurt (Erkaya-Kotan, T. In vitro angiotensin converting enzyme (ACE)-inhibitory and antioxidant activity of probiotic yogurt incorporated with orange fiber during storage. Journal of Food Science and Technology 2020, 57, 2343-2353. https://doi.org/10.1007/s13197-020-04272-1), which thus also explains the reason for the increased acidity of ABT fermented milk fortified with fiber-rich citrus peel. Based on your comment, to clear up the misunderstandings, we also revised the sentence in line 257 in the revised manuscript.
10. Line 249, why moisture content was decreased?
Response: Thank you for your question. Actually, the addition of citrus peel powder increased the total solids in ABT fermented milk and thus caused a decrease in moisture content. The moisture contents in ABT fermented milks with or without citrus peel addition did not change during cold storage.
11. Line 313, better to give the clear discussion on the antimicrobial activity issues
Response: Thank you for your good suggestion. We have revised the discussion on the antimicrobial activity issues in the revised manuscript.
12. Add p-vales in the tables
Response: Thank you for your suggestion. We have added the p value in the table notes of tables in the revised manuscript.
13. line 214, replace “data are presented” as “data presented”
Response: Thank you for your suggestion. We have replaced “data are presented” with “data represent” in the revised manuscript.
Round 2
Reviewer 1 Report
All the comments were addressed scientifically.
Author Response
Thank you for your positive comments again.